# PAH-pollution effects on sensitive and resistant embryos: Integrating structure and function with gene expression

Goran Bozinovic[1,2¤a]*, Damian Shea[1], Zuying Feng[2], David Hinton[3], Tim Sit[4], Marjorie F. Oleksiak[1¤b]

1 Department of Biological Sciences, North Carolina State University, Raleigh, North Carolina, United States of America, 2 Boz Life Science Research and Teaching Institute, San Diego, California, United States of America, 3 Nicholas School of the Environment, Duke University, Durham, North Carolina, United States of America, 4 Department of Entomology and Plant Pathology, North Carolina State University, Raleigh, North Carolina, United States of America

¤a Current address: Division of Biological Sciences, University of California at San Diego, La Jolla, California, United States of America
¤b Current address: Department of Marine Biology and Ecology, Rosenstiel School of Marine and Atmospheric Sciences, University of Miami, Miami, Florida, United States of America
* gbozinovic@ucsd.edu

**Data Availability Statement:** Microarray data have been deposited in NCBI's Gene Expression Omnibus and are accessible through GEO Series accession number GSE130754.

## Abstract

Polycyclic aromatic hydrocarbons (PAHs) are among the most widespread natural and anthropogenic pollutants, and some PAHs are proven developmental toxicants. We chemically characterized clean and heavily polluted sites and exposed fish embryos to PAH polluted sediment extracts during four critical developmental stages. Embryos were collected from *Fundulus heteroclitus* populations inhabiting the clean and heavily polluted Superfund estuary. Embryos of parents from the clean sites are sensitive to PAH pollutants while those of parents from the heavily polluted site are resistant. Chemical analysis of embryos suggests PAH accumulation and pollution-induced toxicity among sensitive embryos during development that ultimately kills all sensitive embryos before hatching, while remarkably, the resistant embryos develop normally. The adverse effects on sensitive embryos are manifested as developmental delays, reduced heart rates, and severe heart, liver, and kidney morphological abnormalities. Gene expression analysis of early somitogenesis, heartbeat initiation, late organogenesis, and pre-hatching developmental stages reveals genes whose expression significantly differs between sensitive and resistant embryo populations and helps to explain mechanisms of sensitivity and resistance to polluted environments during vertebrate animal development.

## Introduction

Embryo development can be particularly sensitive to pollution stress, and developmental abnormalities caused by persistent organic pollutants (POPs) in humans and other species are well documented [1–7]. Embryos and children are more sensitive than adults to POP toxicity

**Funding:** This work was supported by National Institute of Health grants R01 ES011588 and P42 ES007381 to MFO and 2P42 ES010356 to MFO and RD. The funders had no role in study design, data collection and analysis, decision to publish, or preparation of the manuscript.

**Competing interests:** The authors declare that they have no competing interests.

because of their higher cell proliferation rates, lower immunological competence, and decreased ability to detoxify carcinogens and repair DNA damage [8,9]. Epidemiological studies have shown a relationship between perinatal POP exposure and neurological and behavioral disturbances in infants and children [10–13]. For example, ambient POP exposure has been associated with altered physical growth, immune function, and thyroid hormone function in infants [14].

Polycyclic aromatic hydrocarbons (PAHs), which have pyrogenic (combustion derived), biogenic (organism derived) and petrogenic (petroleum derived) sources, are among the most widespread classes of natural and anthropogenic POPs. A number of PAHs, such as benzo[a] pyrene (BaP), are human mutagens and carcinogens, and some are transplacental carcinogens in experimental bioassays, causing tumor formation in liver, lung, lymphatic tissues and nervous system of the offspring [15,16]. Some PAHs also are developmental toxicants in humans [9,17,18]. For instance, mothers with higher ambient air pollution exposure had increased PAH-DNA adducts in their umbilical cords, and their children had significantly decreased weight, length, and head circumference at birth [19], higher anxiety and depression symptoms at 4.8 years, and attention problems at 4.8 and 7 years, suggesting that PAH exposure may adversely affect child behavior [20].

Many PAHs accumulate in soils and sediments and can adversely affect estuarine and coastal marine ecosystems at high enough concentrations. One example of a PAH polluted estuarine ecosystem is the Elizabeth River, VA Superfund site (Superfund sites are some of the United States most contaminated sites that the Environmental Protection Agency is responsible for cleaning up): sediment PAH concentrations at the Elizabeth River Superfund site are extremely high (2,200 mg/kg dry weight) where creosote historically was used to treat wood for use in the marine environment [21,22]. Despite these high PAH concentrations, a population of the teleost *Fundulus heteroclitus* (*F. heteroclitus*) thrives at this site. However, *F. heteroclitus* embryos from nearby clean environments do not survive when grown on Elizabeth River sediment [23] and when exposed to sediment develop significantly more cardiac abnormalities compared to embryos from the Elizabeth River [24], suggesting that compromised heart development contributes to PAH sensitivity. Furthermore, Elizabeth River *F. heteroclitus* offspring were significantly more resistant to oxidative stress than the reference site offspring and showed upregulated antioxidant defenses, which suggests that upregulated antioxidant defense plays a role in both short-term physiological and heritable resistance mechanisms to environmental pollutants [23]. Thus, as for humans, developmental PAH exposure appears to be embryotoxic and teratogenic to *F. heteroclitus* embryos.

To explore mechanisms of POP sensitivity and resistance in natural populations, we exposed *F. heteroclitus* embryos from the Elizabeth River Superfund population and a nearby, relatively non-polluted population to both polluted and clean sediment extracts. We measured chemical uptake, gene expression, and altered embryo anatomy, morphology, and cardiac physiology during four critical developmental stages: somitogenesis, heartbeat initiation, late organogenesis, and pre-hatching.

## Materials and methods

### Sediment and water collection and analysis

All polluted sediments and water were collected in April 2007 from the southern branch of the Elizabeth River, VA (N 36˚ 10.551'; W 75˚ 56.533') during incoming tide; all clean reference sediments and water were collected Magotha Bay, VA in the same manner (N 36˚ 48.473'; W 76˚ 17.654'). The surface layer, 5 cm deep, was scraped off the sediment, placed in 1 L glass jars, and transported to the lab on ice within 24 hours. The multiple sediment sub-samples

from each site were combined in the laboratory, thoroughly mixed manually, and placed in stainless steel containers. Sediments were maintained at -20˚C prior to extraction. Collected site water was transported in 1 L glass jars without headspace, on ice, and maintained at 4˚C prior to extraction. Fieldwork was completed within publicly available lands. No permission was required for sampling sites access and sample collection.

For extraction, sediments were thawed, mixed with site water at a ratio of 1:1 and shaken for 24 hours in 1 L glass jars. The contents were allowed to settle for 6 hours, and overlying water was transferred to 250 ml centrifuge bottles and spun at 1000g. The supernatant was transferred to 50 ml centrifuge tubes and spun again at 3,000 RPM 1000g. The supernatant was serially vacuum-filtered using 3, 1, 0.8, and 0.45 µl Whatman uniprep filter (PFTE), and the extract was stored in 50 ml polypropylene centrifuge tubes at -20˚C prior to embryo exposure and chemical analysis. For chemical analysis, sample volumes were determined, and a surrogate internal standard (SIS) was added. Samples were sequentially extracted three times in a separatory funnel with 50 ml dichloromethane (DCM). After each shaking, the phases were allowed to separate, and the DCM layer was drained through anhydrous sodium sulfate. Fresh DCM was added each time. DCM extracts were combined, concentrated using rotary and nitrogen evaporation techniques, transferred to an autosampler vial and further concentrated to 0.5 ml. An internal standard was added to the sample prior to analysis.

PAHs were analyzed using an Agilent 6890 gas chromatograph (GC) connected to an Agilent 5973 mass selective detector (MSD) and operated in Select Ion Monitoring (SIM) mode. Analytes were separated on a Restek Rtx-5MS column (30m x 250µm dia. x 0.25µm film thickness) with a 5m integrated guard column as described [25]. A complete list of all PAHs detected is available in S1 Table.

## On site chemical analyses

On site chemical exposures were measured with passive sampling devices (PSDs) deployed in aluminum cages for 28 days in early April 2006 at Magotha Bay, VA, and Elizabeth River, VA. The PSDs were made from virgin low-density polyethylene [26,27]. At each site, six samples were placed in the marsh during the incoming low tide approximately 20 m apart. Upon retrieval, the PSDs were wrapped in combusted aluminum foil, placed in a plastic bag, and maintained on ice until frozen. PSDs were transported to the laboratory, where they were maintained frozen at -20˚C until time of analysis. PSDs were analyzed using established methods [26–29].

## Fish maintenance and embryo culture

Adult *F. heteroclitus* were captured from the Elizabeth River, VA (polluted Superfund site) and Magotha, VA (clean, reference site) by minnow traps in April 2007 and transported under controlled temperature and aeration condition to the NCSU Aquatic Laboratory. Fish were acclimated to common conditions of 20˚C and 15 ppt salinity in 40-gal flow-through re-circulating aquatic system tanks for 4 months prior to laboratory spawning. Effluent from the tanks was passed through an activated charcoal filter system and 20% water was changed weekly. Tanks were maintained, and fish were fed (brine shrimp flake, blood meal flake, and Spirulina flake—FOD, Aquatic Biosystems) daily and monitored for heath. Fish were maintained under a pseudo-summer cycle (8 h dark / 16 h light). Fish collection was completed within publicly available lands. No permission was required for sampling sites access or to collect fish for non-commercial purposes, as *F. heteroclitus* does not have endangered or protected status and does not require collecting permits.

Eggs were stripped from 10 females and fertilized by sperm collected from 10 males from each population, resulting in 10 sensitive (C = clean) and 10 resistant (P = polluted) family sets, each with multiple embryo offspring. Fertilized embryos were maintained in Petri dishes with 15 ppt filtered seawater in a 25˚C environmental chamber under light during the initial two stages of development. Fertilization success was confirmed using a stereo microscope. Embryos that successfully reached 2-cell stage within a predetermined time [30] were used in the sediment extract exposure experiment. Adult fish were housed in the lab for use in further research. Experimental procedures for this study were approved by Institutional Animal Care and Use Committee (IACUC) at North Carolina State University (IACUC assurance A3331-01), and non-surgical tissue sampling and fish embryo culturing and maintenance was approved by the IACUC at North Carolina State University (IACUC assurance A3331-01) and the Duke University (IACUC assurance A3195-01).

## Sediment extract exposure

Sediment extracts were thawed to room temperature and 20 ml aliquots were transferred to 50 ml glass scintillation vials pre-rinsed with DI-water. 10 embryos were placed into each vial. Developing sensitive and resistant embryos exposed to clean and contaminated sediment extracts were maintained in an environmental chamber (818 Low Temperature Illuminated Incubator, Precision Scientific, USA) at 25˚C and 16 h light / 8 h dark cycle. Embryo condition and development were monitored daily by examining representative stages using a dissecting stereo microscope (Nikon SME1500, Japan), and time to stage, % normal *versus* abnormal development, and % mortality were noted. Unfertilized eggs, malformed and/or dead embryos were removed from the population, and times and stages of arrest and abnormal development were recorded accordingly. Criteria for time-to-stage was based on >80% success rate of reference embryos cultured in the vials with 15 ppt filtered seawater as well as established developmental timelines [31,32]. Once the normally developing embryos reached one of four developmental stages (21, 25, 31, or 35), images of embryos were recorded using a Micropublisher 5.0 RTV Camera (QImaging) fitted on the stereo microscope, and embryos were either preserved in neutral buffer-formalin for histopathological analysis (stage 31 only) or placed in prechilled 1.5 ml Eppendorf tubes and snap-frozen at -80˚C for chemical and microarray analyses (stages 21, 25, 31, and 35). In addition, heart rates were recorded at stages 31 and 35.

## Embryo chemistry analysis

At stages 21, 25, 31, and 35, 4 pools of 10 embryos for each of the 4 treatments were prepared for chemical analyses. Embryos were quickly dried by gently rolling them on Kimwipes, placed in chilled 1.5 ml polypropylene tubes, and snap frozen at -80˚C until analysis. Pools of 10 frozen embryos were thawed, washed with isopropanol: DI water (1:1), dried, and weighed. Embryos were then placed in 1.5 ml polypropylene tubes pre-rinsed with water: acetone, 1:1, and 3 mm glass beads were added to the tubes which were then frozen in liquid nitrogen for 2 minutes. Embryos were homogenized in a Silamat s5 shaker for 30s, centrifuged at 1000g and then frozen overnight at -20˚C. Surrogate internal recovery standards were added to the tubes and 1 ml DCM was added to the tubes. The tubes were vortexed for 1 minute, the solvent was transferred to the reservoir of a Whatman Mini-UniPrep filter (PFTE, 0.45 μm), 0.5 ml DCM was added to each tube and tubes were vortexed for 1 minute. Solvent was quantitatively transferred to Whatman Mini-UniPrep filter and extracted to a 1.8 ml autosampler vial.

The sample was transferred to a 50 ml Teflon tube, spiked with surrogate internal standard, and serially extracted three times on a shaker table with a total of 75 ml DCM. Extracts were combined, concentrated, and filtered. Lipids were removed by gel-permeation

chromatography (GPC). The lipid fraction was dried and used for lipid determination. The final extract was concentrated to 0.5 ml and a recovery internal standard was added prior to analysis. PAHs were analyzed by GC-MS as for sediment extracts (above).

## Embryo survival and developmental delays

Fertilization success and resultant embryo progress was monitored twice daily by examining representative stages during pre-determined time periods [31,32] using a dissecting stereo microscope (Nikon SME1500, Japan). Time to stage, normal *versus* abnormal development, and mortality were recorded. Unfertilized eggs, malformed and/or dead embryos were removed from the population, and times and stages of arrest and abnormal development were recorded accordingly. Survival rates were measured within a family of each population and as overall survival rates between populations. Embryos that successfully hatched and survived to stage 40 as free-swimming normal-appearing larvae were considered survivors.

To determine developmental delays to stages 31 and 35 within a population, five embryos from five families were monitored in individual 20-ml scintillation vials. Identification of each stage was determined by scoring embryos at predetermined time-periods for both stages 31 (140 hours post-fertilization) and 35 (212 hours post-fertilization [31,32] using a dissecting stereo microscope (Nikon SME1500, Japan) at 70-80X magnification. Multiple images of developing embryos were taken at different phases of each developmental stage. Images were captured with the Micropublisher 5.0 RTV Camera (QImaging) and catalogued, stored, and analyzed using QCapture Pro imaging software.

## Embryo heart rate

Heart rates were determined during early organogenesis and pre-hatching stages (31 and 35, respectively). Individual embryos were tracked to assure that the heart rate was measured from the same embryo at each stage: embryos were placed in a depression slide under a dissecting stereo microscope for 1 min prior to taking heart rate measurements so that the stressed embryo could reestablish resting heart rate (most *F. heteroclitus* embryos temporarily slow their heart rate due to a sudden change of environment, such as transfer from the Petri dish to a well-lit slide surface). Heart rates of each embryo were established by determining number of beats/30 seconds (preliminary results showed no change in the average heart rate when counts were taken for either 30 secs or 1 min). Only those embryos that developed successfully to stage 31 and to 35 were considered for analysis.

## Embryo morphology as determined by *in vivo* trans-chorionic analysis

Embryos were scored for normal *versus* abnormal development. Deformities included various degrees of incompletely differentiated heart chambers, pericardial edema, cranio-facial alterations, loss of pigmentation, scoliosis, tail shortening, and hemorrhaging. Since heart deformities were found to be the most sensitive and reliable endpoint scored in our previous embryo exposure studies, they were more heavily weighted in deformity scorings. At 140 hours (stage 31) and 168 hours-post-fertilization (hpf; stage 35; [31]), images of three randomly selected embryos from each population at stage were photographed, and any resultant morphological abnormalities were catalogued. Ten embryos from each treatment were randomly selected and subjectively scored treatment-blind twice independently (N = 2) for morphological abnormalities using light microscopy. Embryos were scored for severity of heart deformities (tube heart), pericardial edema, hemorrhaging, craniofacial alterations, tail shortening, and pigment loss. Embryo score was based on a 1–5 scale, 1 representing no deformities, 2-mild, 3-moderate, 4-severe, and 5-extreme, respectively. Non-deformed embryos appeared wrapped

approximately 2/3 around the full circumference of the remaining yolk, and with clearly distinguishable cranial ridges, well-defined dark-pigmented eyes with visible retinas, dark and scattered body pigment, clearly distinct atrial and ventricular cardiac regions, absence of hemorrhaging, and the caudal region approximately 1/3 of the body length beginning at the bilobed urinary bladder [31]. The most severely affected embryos were smaller, with the disproportionally reduced cranium size and diminished distance between eyes, complete loss of cranial ridges, reduction of eye pigmentation, near-complete aggregation and overall reduction of body pigmentation, hemorrhaging along the entire shortened caudal region, and complete loss of cardiac muscle integrity and function: heart was a thin-walled, translucent tube structure "tube heart" with the severely compromised or lacking conducing capacity.

Results for each treatment were represented as an average of the individual scores. While all phenotypes were considered in determining the final score, the heart deformities were found to be the most reference and reliable endpoint used in deformity assessment.

## Embryo histopathology

Histopathology was restricted to late organogenesis by which time organs and their respective tissues were unequivocally recognized and presence of blood cells in specific microvasculature elements and/or pigment or precipitate in fluid-filled spaces signified function. These were also correlated with *in vivo* trans-chorionic analysis as described above. Embryos were fixed in 10% neutral buffered formalin (Richard-Allen Sci Neutral Buffer Concentrate). Following initial 24–48 hrs. storage in fixative, embryos were placed in 30% sucrose, and chorions were punctured to facilitate dehydration, clearing and embedment. Fixed embryos were then transferred to mesh tissue cassettes, fixed overnight, and then processed, by (briefly) dehydration, clearing, and embedment in paraffin.

After embedment, 100-micron paraffin blocks (each containing a single embryo) were aligned, and 10 embryos were positioned within a single paraffin block. Embryos were reinfiltrated in paraffin, oriented, and embedded in larger paraffin blocks. This procedure increased the ability to examine multiple, similarly oriented embryos, in each tissue section. Serial sectioning on the rotary microtome (Leica RM 2135) was done at 5 microns and resultant sections were placed on silanized slides. Embryos were routinely stained with hematoxylin and eosin (H&E). Histological sections were viewed under a Nikon Eclipse E600 microscope (Nikon E600, Japan) and images were taken using Lumenera Infinity 2 (model #2-2C) 2.0-megapixel, 12 fps, CCC color camera (Lumenera; Ottawa, Canada). Digital images were analyzed using Eclipse Net Version 1.16.5 software (Enfield, CT).

## Gene expression

Amplified cDNA sequences for 7,000 genes from *F. heteroclitus* cDNA libraries were spotted onto epoxide slides (Corning) using an inkjet printer (Aj100, ArrayJet, Scotland). Libraries were made from all 40 stages of *F. heteroclitus* development, immediately post-hatch whole larvae, and adult tissues. Each slide contained 4 spatially separated arrays of ~7,000 spots (genes) including controls.

**Embryo RNA isolation, amplification, and labeling.**   Pools of ten frozen embryos collected at four developmental stages [21,30–34] were used for RNA isolation, labeling, and microarray hybridization. Embryo RNA was extracted using a TRIzol buffer (Invitrogen, Carlsbad, CA, USA) followed by purification using the Qiagen RNeasy Mini Kit (Qiagen Inc., Valencia, CA, USA).

Purified RNA was quantified with a spectrophotometer, and RNA quality was assessed by gel electrophoresis. RNA for hybridization was prepared by one round of amplification

(aRNA) using Ambion's Amino Allyl MessageAmp aRNA Kit to form copy template RNA by T7 amplification. Amino-allyl UTP was incorporated into targets during T7 transcription, and resulting amino-allyl aRNA was coupled to Cy3 and Cy5 dyes (GE Healthcare, Piscataway, NJ, USA).

Labeled aRNA samples (2 pmol dye/ul) were hybridized to slides in 10 μl of hybridization buffer (50% formamide buffer, 5x SSPE, 1% sodium dodecyl sulfate, 0.2 mg/ml bovine serum albumin, 1 mg/ml denatured salmon sperm DNA (Sigma), and 1 mg/ml RNAse free poly(A) RNA (Sigma)) for 44 hours at 42˚C. Slides were prepared for hybridization by blocking in 5% ethanolamine, 100 mM Tris pH 7.8, and 0.1% SDS added just before use for 30 minutes at room temperature, washed for one hour in 4x SSC, 0.1% SDS at 50˚C, and then boiled for 2 minutes in distilled water to denature the cDNAs. Resulting 16-bit Tiff Images were quantified using ImaGene® (Biodiscovery, Inc.) spotfinding software. Controls and any gene that did not have at least one individual with a signal greater than the average signal from all herring sperm control spots (nonspecific hybridization signal) plus one standard deviation were removed prior to statistical analyses. In total, 6,789 genes were analyzed.

**Experimental design for microarrays.**   A loop design was used for the microarray hybridizations where each sample is hybridized to 2 arrays using both Cy3 and Cy5 labeled fluorophores [35,36] modified as per Altman [37]. The loop consisted of Cy3 and Cy5 labeled embryo aRNAs from 4 biological pools for four developmental stages (S: S21, S25, S31, and S35) and four different treatments (CC, CP, PC, and PP where the first letter denotes the family, sensitive (C = clean) or resistant (P = polluted) and the second letter denotes the exposure, clean (C) or polluted (P) sediment extract. In total, 64 biological pools were hybridized to 32 microarrays. Each array had different combinations of biological pools, so that the most direct comparisons (i.e., PP vs CC stage 31 samples) are hybridized to the same array. The loop formed was S21-PP → S21-CC → S21- PC → S21-CP → S25-PP → S25-CC → S25-PC → S25-CP → S31-PP → S31CC → S31-PC → S31-CP → S35-PP → S35-CC → S35-PC → S35-CP, where each arrow represents a separate hybridization (array) with the biological pool at the base of the arrow labeled with Cy3 and the biological pool at the head of the arrow labeled with Cy5.

## Statistical analyses

**Survival, heart rate, developmental delays, and morphology.**   Differences in the embryo survival, heat rate, time to stage, and morphology among 2 embryo populations and 2 treatments were analyzed with GraphPad Prism (version 9.0.2) using one-way Analysis of Variance (one-way ANOVA, $p < 0.05$) and Bonferroni post-test analysis ($p < 0.01$); pairwise t-test was used to test the differences of means between treatment groups, while Dunnett's one-tailed t-test was used to evaluate differences between "clean" and "polluted" embryos, respectively. Correlation between developmental delays, heart rate, and morphology were determined using non-linear fit polynomial second-order analysis.

**Microarrays.**   $Log_2$ measures of gene expression were normalized using a linear mixed model in JMP Genomics (version 9.1) to remove the effects of dye (fixed effect) and array (random effect) following a joint regional and spatial Lowess transformation in MAANOVA Version 0.98.8 for R to account for both intensity and spatial bias [38].

The model was of the form $y_{ij} = \mu + A_i + D_j + (A \times D)_{ij} + e_{ij}$, where, $y_{ij}$ is the signal from the $i^{th}$ array with dye j, $\mu$ is the sample mean, $A_i$ and $D_j$ are the overall variation in arrays and dyes (Cy3 and Cy5), $(A \times D)_{ij}$ is the array-by-dye interaction and $e_{ij}$ is the stochastic error [39,40].

Residuals from above model were used for a gene-by-gene analyses using stage, treatment, population, population-by-treatment, stage-by-treatment, population-by-stage, stage-by-

treatment-by-population and dye as fixed effects. The model was $r_{ijknm} = \mu + A_i + D_j + T_k + P_n + S_m + (T \times P)_{nk} + (T \times S)_{km} + (S \times P)_{mn} + (S \times T \times P)_{mkn} + e_{ijknm}$ where $T_k$ is the $k^{th}$ treatment (sediment extract), $P_n$ is the $n^{th}$ population, $S_m$ is the $m^{th}$ stage, $(T \times P)_{nk}$ is treatment-by-population interaction, $(T \times S)_{km}$ is the treatment-by-stage interaction, $(S \times P)_{mn}$ is the stage-by-population interaction, and $(S \times T \times P)_{mkn}$ is the stage-by-treatment-by-population interaction.

Because we were interested in treatment and population effects more than stage effects, we also analyzed the data-by-stage. Here the model was $r_{ijkn} = \mu + A_i + D_j + T_k + P_n + (T \times P)_{nk} + e_{ijkn}$ where $T_k$ is the $k^{th}$ treatment (polluted or reference sediment extract), $P_n$ is the $n^{th}$ population, and $(T \times P)_{nk}$ is treatment-by-population interaction.

For all mixed model analyses, we used a nominal p-value cut-off for significant genes of $p < 0.01$. Using this p-value reveals more genes that may be differentially expressed but risks identifying genes that may be false positives. False discovery rate (pFDR $< 0.05$) was also used for filtering residuals.

Hierarchical clustering used JMP Genomics (version 9.1) Ward method with Center rows [41].

## Results

### Sediment extract chemistry

Compared to the reference site, the Elizabeth River sediment extract was highly polluted with a variety of PAHs, including both low (less than four aromatic rings) and high (four or more aromatic rings) molecular weight compounds (Fig 1A, S1 Table) [42,43]. The only chemicals detected in the reference site extract were naphthalene (0.8% of the Elizabeth River level) and 1- and 2-methylnaphthalenes (4% of Elizabeth River levels).

### Passive sampling devices (PSDs)

PSDs deployed at sediment-water interface at the reference site showed very low concentrations of only 10 of the 42 PAHs for which we tested. These were mostly low molecular weight naphthalenes (Fig 1B, S1 Table). The highest PAH concentration was 11.86 ng/L C1-naphthalenes, followed by 7.86 ng/L fluoranthene. In contrast, PSDs deployed at the Elizabeth River site showed all 42 tested PAHs, with the lowest concentration of 9.2 ng/L acenaphthylene and 11.5 ng/L biphenyl and high concentrations among high molecular weight PAHs– 690 ng/L phenanthrene, 2,075 ng/L pyrene, and 7,048 ng/L fluoranthene.

### Embryo chemistry analysis

Embryo PAH content during four critical developmental stages was measured to examine the protective role of the chorion, the outermost embryonic membrane. We detected the highest PAH concentrations in sensitive embryos exposed to polluted sediment extracts while PAH concentrations in sensitive embryos developing in clean extracts were the lowest among the four treatment groups. Nine PAHs were detected in sensitive embryos developing in polluted extracts and resistant embryos developing in both clean and polluted extract; naphthalenes (C1-C4) were the most prevalent PAHs detected in all treatment groups (Fig 1C, a-d). PAH concentrations of sensitive and resistant embryos in polluted extracts were similar during later developmental stages.

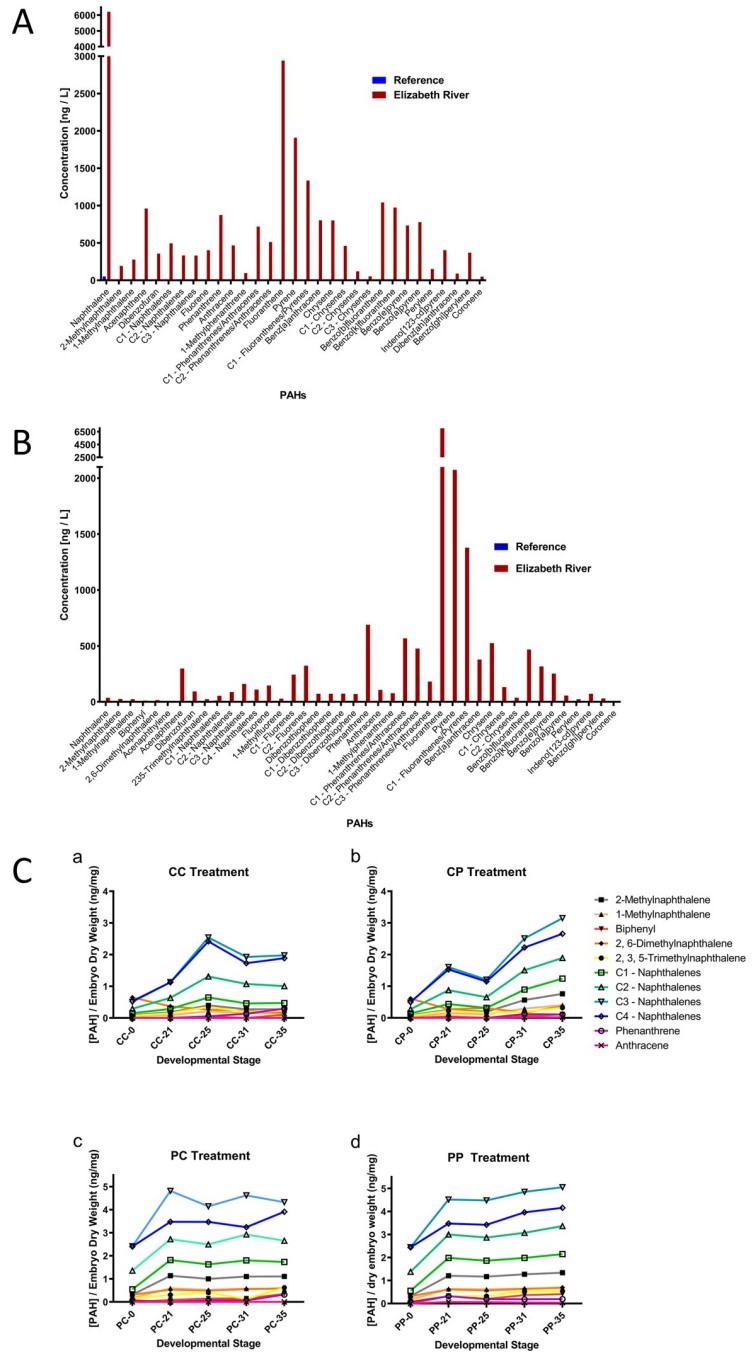

**Fig 1. Chemical assessment of clean and polluted sediments, and sensitive and resistant embryos after exposures.**
(A) Chemical Analyses of PAH concentrations present in Elizabeth River, VA, and Magotha Bay, VA sediment extracts. Compared to the reference site, Elizabeth River sediment extract is highly polluted with both low and high molecular weight of PAHs, relevant to reference site. The only PAH detected in the reference site extract are naphthalene (0.8% of the Elizabeth River level) and 1 and 2-methylnaphthalenes (4% of Elizabeth River levels). (B) PAH-exposure analysis using PSD at Elizabeth River, VA, and Magotha Bay, VA sites. Low concentrations of only 10 of the 42 PAHs tested, mostly low molecular weight naphthalenes, are detected at the reference site: C1—Naphthalenes have the highest concentration at 11.86 ng/L, while other detected PAHs are all below 10 ng/L. All 42 tested PAHs are detected the Elizabeth River site, with the high concentrations among high molecular weight PAHs– 689.57 ng/L phenanthrene, 2,075.03 ng/L pyrene, and 7,048.59 ng/L fluoranthene. (C) Chemical Analysis of sensitive and resistant embryos exposed to clean and polluted sediments extracts during 4 developmental stages (a–CC: sensitive embryos in clean extracts; b–CP: sensitive embryos in polluted extracts; c–PC: resistant embryos in clean extracts; d–PP: resistant embryo in polluted extracts). The highest concentrations of PAHs are detected in clean embryos exposed to polluted

sediments; PAH concentrations in clean embryos developing in clean extracts were the lowest among the four treatment groups.

## Embryo survival and developmental delays

We found significant survival rate differences among treatment groups (Fig 2A, one-way ANOVA, p < 0.05). Sensitive embryos developing in clean sediment extracts during stages 31 (late organogenesis), 35 (pre-hatching), and hatching had the highest survival rates (87.5%) among all treatment groups. Resistant embryos raised in both clean and polluted sediment extracts (PC and PP, where the first letter shows the parental population, polluted or clean, and the second letter shows the treatment, exposure to polluted or clean sediment extract) and sensitive embryos raised on polluted extracts (CP) had significantly lower survival rates (57.5–65%). Importantly, none of the sensitive embryos exposed to polluted sediment extracts hatched; they subsequently died (Bonferroni post-test: CC hatch vs. CO hatch: p < 0.01, t = 12.6). Although the survival of embryos at stage 35 did not statistically differ among treatments, hatching success was significantly lower among PC and PP embryos compared to sensitive embryos developing in clean sediment extracts (CC hatch vs. PC or PP hatch: p < 0.05, t = 4.171).

Significant delays (Fig 2B, one-way ANOVA, p < 0.01) and incomplete development were noted among sensitive embryos exposed to polluted extract (CP) at both stages 31 (Bonferroni post-test, CC31 vs. CP31: p < 0.01, t = 5.542) and 35 (CC35 vs. CP35, p < 0.01, t = 5.924), with slightly more pronounced delays in the latter stage when compared to other treatment groups (Fig 2B). Although slightly delayed, resistant embryo development did not differ significantly, and most of the embryos developed on time, including the resistant embryos exposed to polluted extracts. While CC, PC, and PP embryos reached pre-hatching stage 35 within the expected time (approximately 212 hours), the CP were, on average, 3 stages (52 hours) behind (Fig 2B and 2C, a).

## Embryo heart rates

The average heart rates among sensitive embryos developing in clean extracts (CC) during stages 31 and 35 were 125.2 and 125.8 bpm, respectively, which were significantly higher than the rates observed among resistant embryos in clean sediment extracts (CP; Fig 2D; one-way ANOVA, p < 0.01; Bonferroni post-test: stage 31, p < 0.05, t = 7.14; stage 35, p < 0.01, t = 5.84). We measured the lowest average heart rate in CP embryos during stage 31 (108.7 bpm). This rate increased slightly during stage 35 (112.3 bpm). Average heart rates during both stages in PC (120.0 bpm; 122.3 bpm) and PP (124.0 bpm; 123.8 bpm) embryos did not differ significantly either from CC embryos or each other, suggesting no exposure effects on heart rate.

## Embryo morphology

Representative morphological deformities are presented in Fig 3A, a-d. Sensitive embryos raised on polluted sediment (CP) were significantly deformed when compared to all other treatment groups (Fig 3B; one-way ANOVA, p < 0.01; Bonferroni post-test: CC31 vs. CP31, p < 0.01, t = 12.83; CC35 vs. CP35, p < 0.01, t = 12.15). The most severe morphological abnormalities were noted among all sensitive embryos exposed to polluted sediment extracts (N = 12 / treatment group). These deformities included pericardial edema, hemorrhaging, cranio-facial malformations [44], tail shortening, hemorrhaging, and general pigment loss (Fig

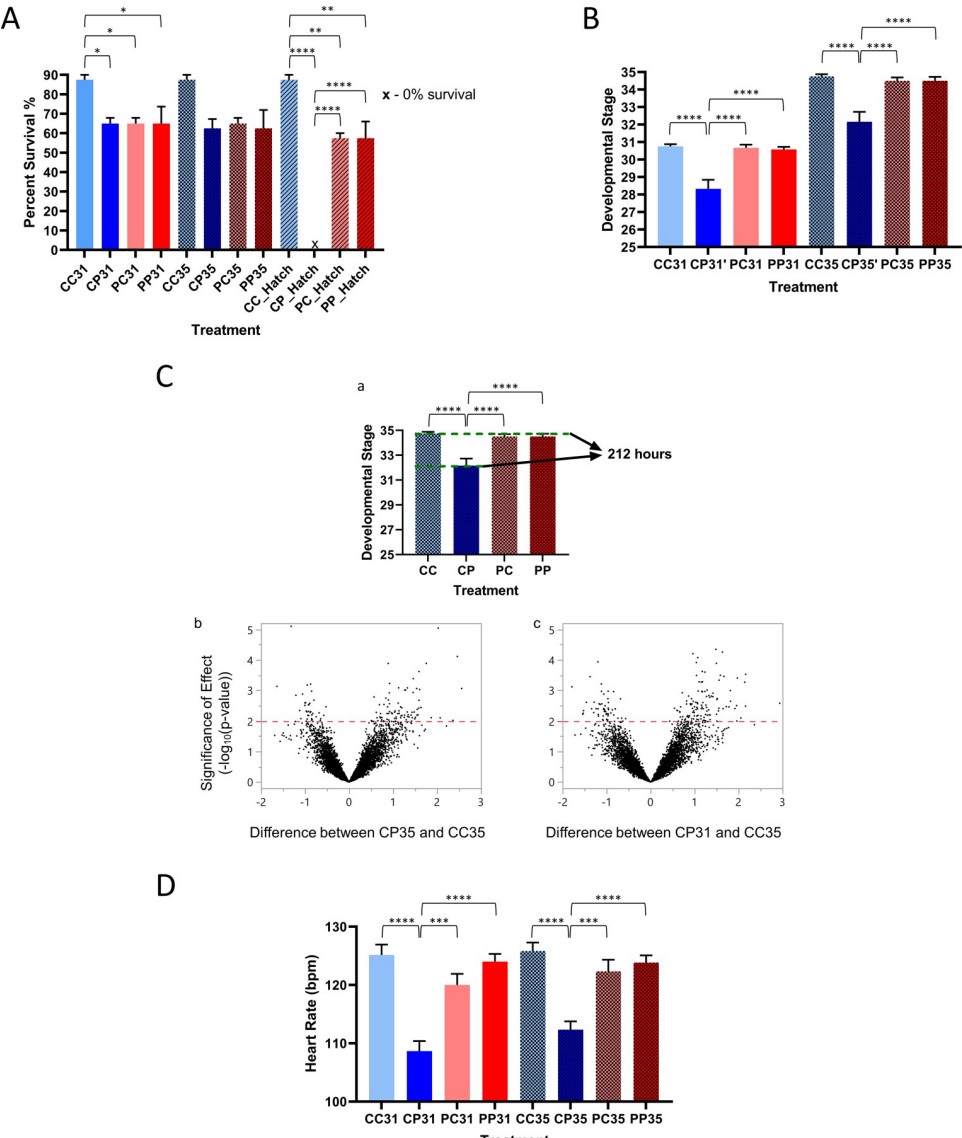

**Fig 2. Survival, developmental delays, and heart rates of sensitive and resistant embryos developing in clean and polluted sediment extracts.** (A) Survival rates of sensitive and resistant embryos exposed to clean and polluted sediment extracts at stages 31 and 35. Survival rates are significantly different among treatment groups (one-way ANOVA; Bonferroni post-test, p < 0.05). Sensitive embryos developing in clean sediment extracts during stages 31, 35, and hatching, had the highest survival rates (87.5%). Resistant embryos exposed to clean and polluted sediment extracts (PC and PP) and sensitive embryos exposed to polluted extracts (CP) had significantly lower survival rates (57.5–65%). (B) Time-to-stage for embryos from two populations developing in clean and polluted sediment extracts. Significantly slower development of sensitive embryos exposed to polluted sediment extracts is noted for both stages 31 and 35 (one-way ANOVA; Bonferroni post-test, p < 0.01). (C) a–Developmental delays of sensitive embryos exposed to polluted sediment extracts (CP) before hatching (stage 35). CC, PC, and PP embryos reached pre-hatching stage 35 within the expected time (approximately 212 hours), while the CP embryos were 3 stages (52 hours) behind. b– Pairwise differences in gene expression between CP and CC embryos at stage 35 based on the anatomical and morphological similarities and not the time-to-stage. c–Pairwise differences between CP embryos at stage 31 and CC embryos at stage 35 considering development of two treatment groups at the same time frame only. Significances of differences as -log$_{10}$(p-values) are plotted against log2 differences in expression of adjacent stages. -log$_{10}$(p-values) range from 0 to 5 and log2 differences in expression range from -2 to 3 (-4-fold to 8-fold differences in expression). Distribution of differentially expressed genes (p < 0.01) between the developmental stages of treatment groups are shown. (D) Heart rates (bpm) of sensitive and resistant embryos. Significantly lower heart rates are detected among sensitive embryos developing in contaminated sediment extracts and resistant embryos developing in clean sediment extracts (one-way ANOVA; Bonferroni post-test, p<0.01). *, p < 0.05; **, p < 0.01; ***, p < 0.001; ****, p < 0.0001.

3A, a-d). The most deformed embryo hearts (score $\geq$ 4 on a scale of 1 to 5) failed to differentiate, resulting in a "tube-heart" structure, which appeared as a barely visible long tube with transparent fluid slowly trickling through (instead of a 2 fully-formed round chambers with the red blood forcefully pumping from the atrium into the ventricle, as seen in the sensitive embryos). The average morphological score for sensitive embryos raised on clean extracts (CC) was 1.083 for both stages 31 and 35 (Fig 3B), indicating that almost all the reference embryos developed normally. Except for the CP treatment group, there were no significant differences in morphology scores between CC embryos and other treatment groups. Significantly deformed CP embryos were either severely or extremely deformed (score 4 and 5) due to exposure to polluted ER sediment extracts, with average scores of 4.083 (stage 31), and 4.250 (stage 35).

Morphology scores, heart rate, and stage delays were correlated among sensitive embryos exposed to PAH-polluted sediment extracts while there was no correlation among resistant embryos exposed to either clean or polluted sediment extracts (Fig 3C, a-c). As sensitive embryos became more deformed and their morphology scores increased, their heart rates significantly decreased ($R^2$ = 0.78; Fig 3C, bi). The morphological deformities also caused significant development delays, characterized by altered heart rate ($R^2$ morphology—developmental delays = 0.82, Fig 3C, ai; $R^2$ developmental delays–heart rate = 0.65, Fig 3C, ci). By stage 35, sensitive embryos exposed to polluted sediments were 3 stages (48 hours) behind, relative to other treatment groups (Fig 2C). There was no correlation between developmental delays, heart rate, and morphology scores among resistant embryos (Fig 3C, aii, bii, and cii).

## Embryo histopathology

Histological abnormalities were detected only in sensitive embryos exposed to polluted sediments. Due to difficulties in sectioning chorionated (i.e., unhatched) embryos, we were not able to analyze all individuals. Among the informative histology slides, we observed the most prevalent alterations in heart development (70% of embryos, N = 8) and minor alterations in liver and kidney tissues (40%, N = 5). Representative histological sections of stage 31 sensitive embryos exposed to polluted sediment extracts are presented in Fig 3D, a-i.

**a-b: Liver and kidney (40x).** Fig 3D, a and b show a portion of the foregut, liver, and adjacent renal kidney tubules. The foregut epithelium appears essentially normal and mitotic activity was noted in several of the epithelial cells. The liver was comprised of hepatocytes, biliary epithelial cells, sinusoid endothelial cells, and fat storing cells of Ito. Thus, cellular differentiation of the liver, as found within an older organism, was present. The cytoplasm of the hepatocytes was stained throughout, and numerous hepatocytes contained small round vacuoles suggesting lipids. This contrasts with the glycogen-laden hepatocytes of livers from resistant embryos raised over clean sediment. Numerous mitotic figures were seen in the hepatocyte population signifying continued and expected organ growth. Within the renal tubules, epithelial cells were swollen, and this change was particularly apparent at basal regions of cells; tubule lumens were compromised by the swollen epithelial cells.

**c: Ventricular wall (40x).** Fig 3D, c shows a section through the rostralmost portion of the head. The ventricular wall is at the center of the field. Numerous nucleated red blood cells are found within the ventricle lumen. A large subendothelial separation is apparent in the right-hand portion of the field. This appearance suggests fluid accumulation within the ventricle wall.

**d: Atrium (40x).** Fig 3D, d is an elongated view of the atrium showing nucleated red blood cells within the lumen and subendothelial vacuolation as well as higher degree separation of subendothelial elements (lower left of the vessel wall; see arrows). The atrium wall's integrity is compromised by subendothelial vacuolation and greater diameter separation of

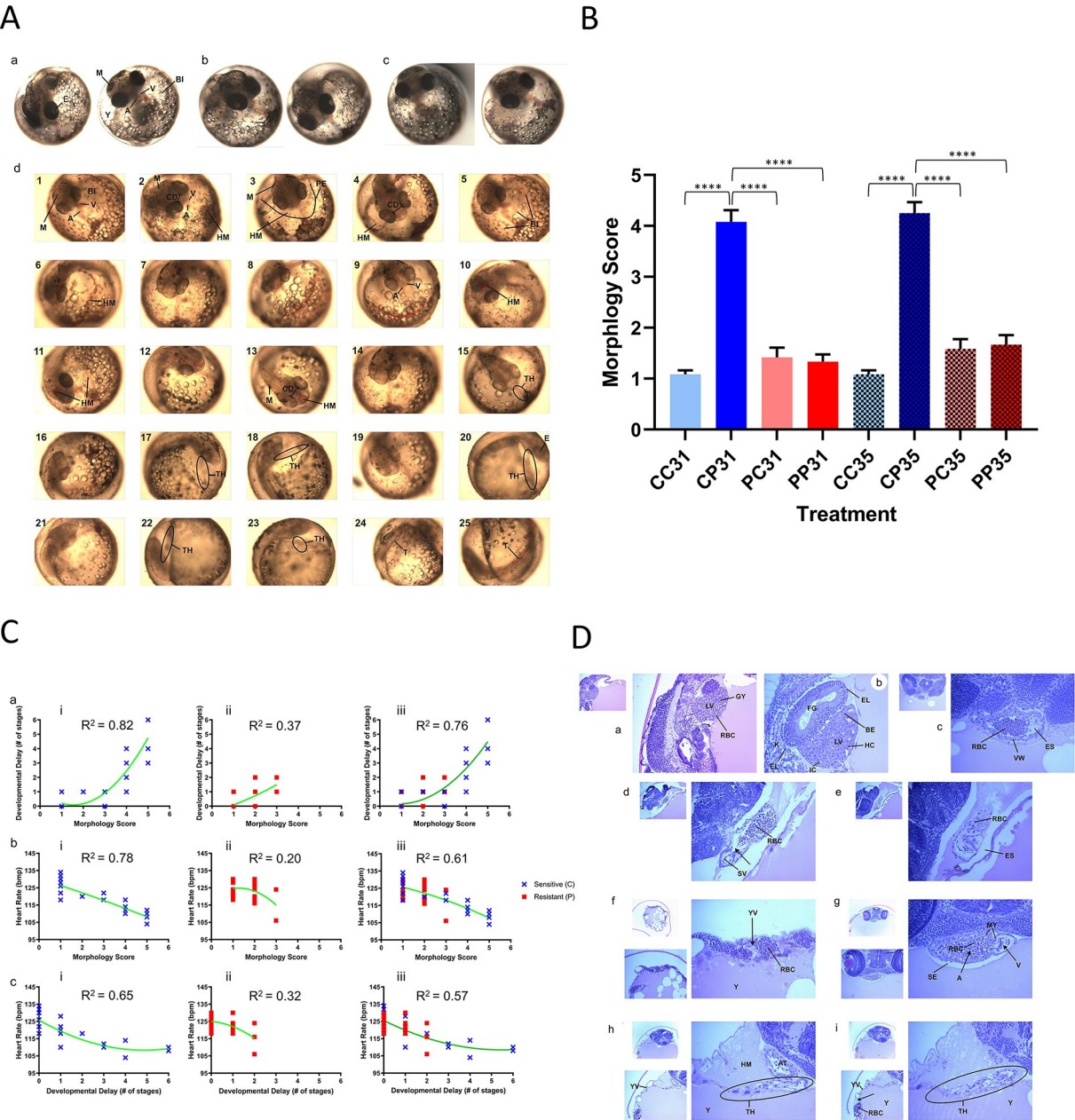

**Fig 3. Embryo morphology and developmental delay–heart rate–morphology correlations of sensitive and resistant embryos exposed to clean and polluted sediment extract.** (A) Embryo Morphology at stage 31. a–normal CC embryo; b–normal PC embryo; c–normal PP emryo; d–progression of deformities among exposed sensitive embryos: 1–5 = mildly deformed (score 2); 6–10 –moderately deformed (score 3); 11–16 –severely deformed (score 4); 17–25 = extremely deformed (score 5). A–atrium; BI–blood island; E–eye; CD–cranial deformity (reduced cranial width and eye distance, diminished cranial ridges); HM–hemorrhage; M–melanin; PE–pericardial edema; T–tail; TH–tube heart; V–ventricle. The most severe morphological abnormalities were noted among all sensitive embryos exposed to polluted sediment extracts (N = 12 / treatment group). (B) Embryo deformity assessment among sensitive (blue) and resistant (red) embryos exposed to clean and polluted sediment extracts. One-way ANOVA (p < 0.01) and Bonferroni post-test (p < 0.01) revealed statistical differences between CP embryos and all other treatment groups. The average morphological score for sensitive embryos raised on clean extracts (CC) was 1.083 for both stages 31 and 35: almost all of the sensitive embryos developed normally. Except for the CP treatment group, there were no significant differences in morphology scores between CC embryos and other treatment groups. Significantly deformed sensitive (CP) embryos were severely or extremely deformed (score 4 and 5) due to exposure to polluted ER sediment extracts, with average scores of 4.083 (stage 31), and 4.250 (stage 35). ****, p < 0.0001. (C) Morphology scores, heart rate, and stage delays are correlated among sensitive embryos exposed to PAH-polluted sediment extracts. Green curves are regression lines indicating the correlation between the specified variables; corresponding $R^2$ values, coefficients of determination, represent the strength of the correlations. a: Developmental Delays–Morphology, b: Heart Rate–Morphology, and c: Heart rate–Developmental Delays correlations among i) sensitive embryos, ii) resistant embryos, iii) both embryo populations. bi) shows as

sensitive embryos become more deformed and their morphology scores increase, their heart rates significantly decreased ($R^2 = 0.78$); ai) shows morphological deformities also caused significant development delays ($R^2 = 0.82$), characterized by ci) altered heart rate ($R^2 = 0.65$). (D) Histopathology of stage 31 sensitive and resistant embryos exposed to clean and polluted sediment extract. Histological abnormalities were detected only in sensitive embryos exposed to contaminated sediments. The most prevalent alterations are observed in heart development (70% of embryos, N = 8) and minor alterations in liver and kidney tissues (40%, N = 5). a, b–Liver and Kidney; c–Ventricular Wall; d–Atrium; e–Tube Heart; f–Yolk Vein Vascular Alterations and Yolk Sac Hemorrhage; g–Atrio-Ventricular Alterations; h, i–Tube Heart. A, AT–atrium, BE–biliary epithelial cells, EI–epithelial cells; ES–effusion; FG–foregut; GY–glycogen; HC–hepatocytes, HM–Hemorrhage, IC–Ito cells, K–kidney, LV–liver, MY–myocytes, RBC–red blood cells, SE—subendothelium, SV–subendothelial vacuolation, TH–tube heart, Y–yolk, YV–yolk vein, VW–vessel wall.

mural elements. The clear space surrounding the atrium is devoid of cells indicating that the response is neither hemorrhage nor inflammation but rather an effusion possibly due to a loss of the chamber wall's integrity.

**e: Tube heart (40x).** This identical embryo imaged *in vivo* (Fig 3D, e) showed an elongated, tubular heart. While the normal heart contraction rate of a stage 31 embryo is between 120 and 132 per minute (bpm), the tubular heart contraction rate in this embryo was 54 bpm. Furthermore, after embryo fixation and processing for high-resolution light microscopy, we could confirm and extend *in vivo* observations. In this view, the section passes through a large vein near the atrium. The endothelium of this vessel is widely separated from outlying structures by what appears to be a fluid-filled space. The absence of cells within this space suggests that the process is neither hemorrhage nor inflammation, rather effusion. While it is tempting to regard the eosinophilic proteinaceous material surrounding the vessel profile as sign of an exudate, note its high similarity to yolk and the fact that we see discrete red round globules in each.

**f: Yolk vein vascular alterations (10x) and yolk sac hemorrhage (40x).** The associated alterations in vein, atrium, and ventricle walls were also noted in the yolk veins (Fig 3D, f). Large clear artifacts of preparation are seen in yolk material. Note the large accumulations of nucleated red blood cells around the yolk sac periphery in this specimen. Higher magnification vascular alterations from the previous figure reveal an absence of vessel wall and a large portion of the circumference devoid of endothelium. This hemorrhage of yolk sac veins was often associated with the presence of a tubular heart. *In vivo* imaging of this same individual showed hemorrhage into the yolk sac.

**g: Atrio-ventricular alterations (40x).** Fig 3D, g illustrates the elements found at the juncture of the atrium and a ventricle in the stage 31 embryo. The atrium is shown at the right of the figure, and a large subendothelial space has formed separating the endothelium from the outlying atrial myocytes. Changes within the ventricular wall appear restricted to myocyte vacuolation.

**h: Tube heart (10x).** Toward the bottom left of the field a group of nucleated red blood cells is shown; higher magnification view of this area revealed (not shown in figures) an absence of endothelial lining for this extent of the yolk vein (Fig 3D, h). Near the top of the field, the vessel extends to near the atrium. At this point analysis of the lumen and wall of the vessels reveals a paucity of red blood cells. During *in vivo* observations of this identical embryo, the lumen of the elongated tube heart was described as lacking pigmentation (showing clear fluid) normally associated with large vessels of embryos and having a very weak and slowed contraction rate.

**i: Tube heart (20x).** Fig 3D, i includes an area of hemorrhage in the lower left of the field and is followed by, as one moves up in the field and to the right, profiles of subsequent portions of the yolk vein. This figure confirms and extends observations made through the dissection microscope (*in vivo* imaging). The arrows indicate portions of the yolk vein, which are devoid of red blood cells. The slowly contracting tubular heart contained fluid that lacked the normal

**Table 1. Symbols used in reporting results between two embryos populations exposed to clean and polluted sediment extracts.**

| | Treatment | |
|---|---|---|
| **Embryo Population** | **Clean Sediment Extract = C$^2$** | **Polluted Sediment Extract = P$^2$** |
| Magotha–Sensitive = C$^1$ | C$^1$C$^2$ = **CC** | C$^1$P$^2$ = **CP** |
| Elizabeth River–Resistant = P$^1$ | P$^1$C$^2$ = **PC** | P$^1$P$^2$ = **PP** |

Two *F. heteroclitus* embryo populations–sensitive (C1) and resistant (P1) were exposed to clean reference site sediment extracts from Magotha Bay, VA (C2) and PAH-polluted sediment extracts from Elizabeth River, VA (P2), resulting in four embryo treatment groups. Embryos' development–PAH content, time-to-stage, heart rates, morphology, and gene expression were analyzed during four developmental stage: Early somitogenesis (21), heart differentiation (25), organodifferentiation (31), and pre-hatching (35).

pigmentation of a large vein in stage 31 embryos. Perhaps the lack of pigment is due to hemorrhage involving afferent veins that lead to the tubular heart.

## Gene expression

We analyzed expression of 6,789 *F. heteroclitus* genes during four critical developmental stages of two *F. heteroclitus* embryo populations. We exposed sensitive embryos of parents from the clean site and resistant embryos of parents from the polluted site to both clean and polluted sediment extracts (Table 1) during early somitogenesis (stage 21), heartbeat initiation (stage 25), late organogenesis (stage 31), and pre-hatching (stage 35). In total, 1178 genes (17.4% of 6,789; 262 genes at pFDR < 0.05) were significantly differently expressed (mixed model ANOVA, p < 0.01, Table 2). 114 genes (1.7% of 6,789; 36 genes at pFDR < 0.05) show a significant treatment-by-population-by-stage interaction. Hierarchical clustering of these 114 genes (Fig 4A) reveals four main clusters among treatment groups based on gene expression pattern similarities.

The numbers of significant gene expression differences for stages 21, 25, 31 and 35 are shown in Table 3. The highest total number of significant gene expression differences due to population differences and population-by-treatment effects is at stage 31 (90 genes for both population and population-by-treatment), while the highest numbers of significant gene expression differences due to treatment alone is at stage 35 (232 genes), followed by stage 31 (157 genes). The lowest number of significant gene expression differences due to population only (24 genes), and treatment alone (21 genes), occurs at stage 25, while the lowest number of significant gene expression differences due to population-by-treatment interaction is at stage 35 (21 genes). The highest total number of significant gene expression differences (308) is at stage 35, while the lowest number (77), is at stage 25 (Table 3, Fig 4B).

**Table 2. Number and percent of significant genes (mixed model ANOVA, p < 0.01 and pFDR < 0.05) of 6,789 analyzed due to population, treatment, and stage alone, and population-by-treatment, population-by-stage, treatment-by-stage, and population-by-treatment-by-stage interactions.**

| | Population | Treatment | Stage | Population X Treatment | Population X Stage | Treatment X Stage | Population X Treatment X Stage |
|---|---|---|---|---|---|---|---|
| **# Sig. Genes (p < 0.01)** | 85 | 270 | 367 | 431 | 108 | 118 | 114 |
| **% Sig. Genes (p < 0.01)** | 1.25 | 3.98 | 5.41 | 6.35 | 1.59 | 1.74 | 1.68 |
| **# Sig. Genes pFDR (p < 0.05)** | 34 | 63 | 196 | 53 | 39 | 59 | 36 |

17.35% (1,178 genes) were significantly differently expressed (mixed model ANOVA, p < 0.01).

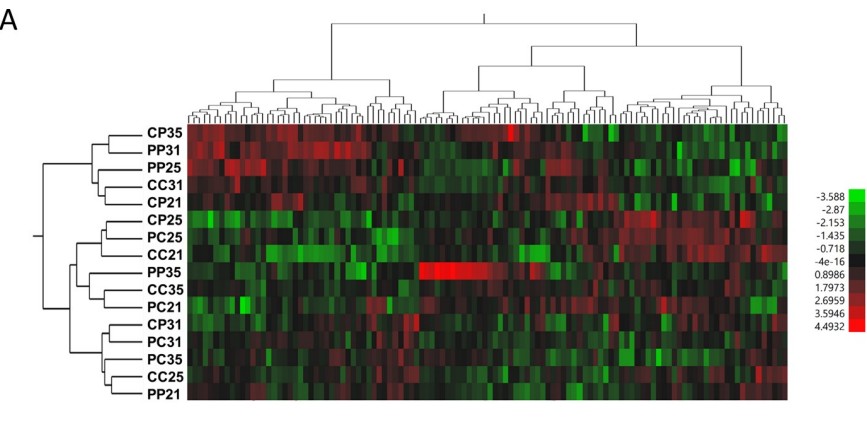

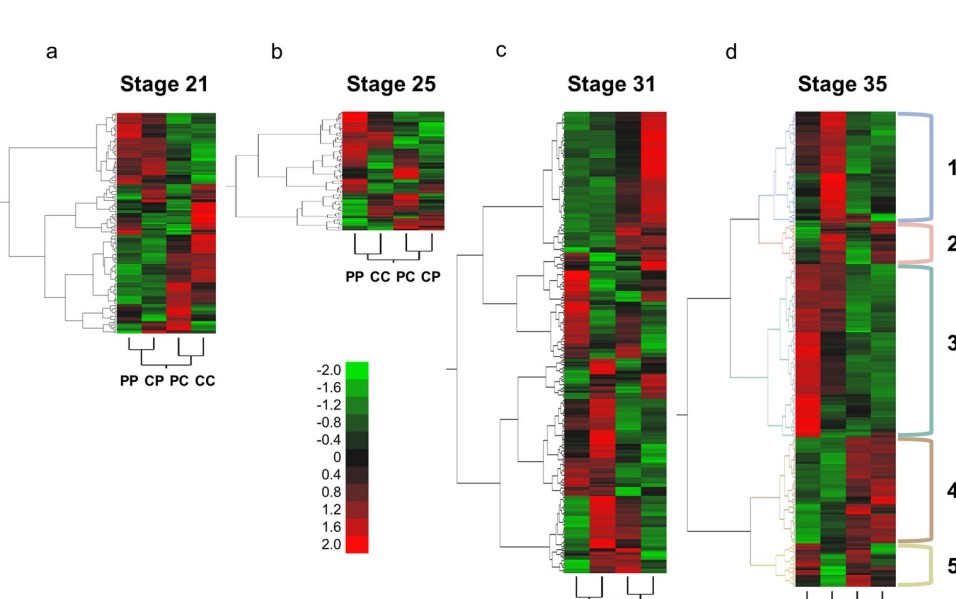

**Fig 4. Differentially expressed genes (p < 0.01) during four critical developmental stages among sensitive and resistant embryo populations exposed to clean and polluted sediment.** Each row represents the embryo, treatment, and developmental stage; each column represents a gene. Clusters of genes with similar expression patterns relative to embryo treatment groups and developmental stage are shown on the left (gene tree). (A) Heat map and hierarchical clustering of significant genes (mixed model ANOVA, p < 0.01) in treatment-by-population-by-stage analysis of sensitive and resistant embryos exposed to clean and polluted sediment extracts. Expression of 6,789 genes was analyzed during four critical developmental stages—early somitogenesis (stage 21), heartbeat initiation (stage 25), late organogenesis (stage 31), and prehatching (stage 35)—of two *F. heteroclitus* embryo populations. Hierarchical clustering of the 114 genes with a significant treatment-by-population-by-stage interaction reveals four main clusters among treatment groups based on the similarities in gene expression patterns. Red indicates high expression levels and green represents low expression levels. (B) Heat maps and hierarchical clustering of genes whose expression significantly differs (pairwise comparisons, p < 0.01) within each developmental stage among sensitive and resistant embryos exposed to clean and polluted sediment extracts (CC, CP, PC, PP). The highest total number of significantly differently expressed genes (308) is at stage 35, while the lowest number of significantly differently expressed genes (77), is at stage 25. Red indicates high expression levels and green represents low expression levels. Cluster labeling shows five major clusters (highlighted by colors) of Stage 35 analyzed in the Discussion section.

Significances of differences as -log10(p-values) are plotted against $\log_2$ differences in expression of adjacent stages. $-\log_{10}$(p-values) range from 0 to 6, and $\log_2$ differences in expression range from -2 to 3 (-4-fold to 8-fold differences in expression). Considering only

**Table 3. Number of significant genes due to stage, treatment, and treatment-by-stage interactions stages 21, 25, 31, and 35 of *F. heroclitus* embryo development.**

| Stage | Population | Treatment | Population X Treatment | Total |
|-------|-----------|-----------|-----------------------|-------|
| 21 | 28 | 93 | 24 | 143 |
| 25 | 24 | 21 | 32 | 77 |
| 31 | 72 | 157 | 90 | 299 |
| 35 | 83 | 232 | 21 | 308 |

The highest total number of significant genes due to population-by-treatment effects is at stage 31 (90 genes), while the highest numbers of significant genes due to population alone and treatment alone is at stage 35 (83 genes for population, and 232 genes for treatment).

anatomical and morphological embryo similarities of the two treatment groups, we analyzed pairwise gene expression difference between sensitive embryos exposed to PAH-contaminated sediments (CP) at stage 35 and sensitive embryos developing in clean water (CC) at stage 35 (Fig 2C, b). 49 genes have >2-fold higher expression among CP embryos and 9 genes have >2-fold higher expression in CC embryos. Considering only the time as the developmental reference (in this case 212 hours, since >80% of reference embryos reached the pre-hatching stage; Fig 2C, c), 74 genes have >2-fold higher expression among CP embryos and 32 genes have >2-fold higher expression among CC embryos.

The hierarchical clustering of significant genes based on mixed model analyses at each stage is presented in Fig 4B. Gene expression pattern similarities reveal that at stages 21, 31, and 35, both embryo populations group-by-treatment (CP with PP, and CC with PC), while at stage 25, sensitive embryos from polluted sediments groups with resistant embryos from clean sediments (CP with PC), and sensitive embryos from clean sediments group with resistant embryos from polluted sediments (CC with PP).

## Discussion

Our previous *F. heteroclitus* embryo work provided anatomical guidelines and statistical support for the temporal dynamics of developmental gene expression during all vertebrate developmental stages [30]. Furthermore, gene expression analyses of unexposed embryos from sensitive and resistant populations found relatively few gene expression differences suggesting that developmental canalization might drive gene expression and biologically important differences between sensitive and resistant embryos might only manifest with exposure, i.e., are dependent on gene by environment interactions [45]. However, the phenotypic changes and large morphological differences exhibited by sensitive and resistant populations exposed to specific PAH toxicants pollutants [46] were accompanied by significant gene expression changes of relatively few genes. In this study, we exposed sensitive and resistant *F. heteroclitus* populations to clean and polluted sediment extract to relate exposure with morphology, histology, cardiac physiology, and gene expression during four critical developmental stages.

Chemical analysis of the embryos' parental environments shows conspicuous differences in PAH concentrations of sediment extracts between the relative unpolluted reference site (Magotha Bay, VA) and the polluted Elizabeth River, VA site (Fig 1A). The Elizabeth River sediments are heavily polluted with numerous low and high-molecular weight PAHs, which have been shown to elicit toxicological effects during *F. heteroclitus* embryogenesis [23,47,48]. In contrast, we detected only three low molecular weight PAHs in Magotha Bay sediments, indicating that the parents of embryos sensitive to PAHs live in a relatively clean environment as compared to fish from the highly PAH-polluted Elizabeth River site. We also measured PAH concentrations at these two sites using passive sampling devices (PSDs), which passively accumulate the dissolved and bioavailable fraction of organic chemicals and are used to

estimate timeweighted average exposure to PAHs. These membranes are not susceptible to biological variation affecting bioaccumulation in living organisms and correlate well with PAH accumulation in caged mussels [26]. We detected high concentrations of all 42 PAHs tested in Elizabeth River sediment extracts compared to only ten low-concentration PAHs detected in the reference site sediment extract, indicating differences in exposures to *F. hetero-clitus* adults and embryos at these two sites (Fig 1B).

To determine the concentrations and species of PAHs that penetrate the embryos' protective membrane (chorion) and come in direct contact with the developing embryo, we measured unexposed embryos at the two-cell stage and embryos exposed to sediment extracts from the clean and polluted sites at four critical developmental stages: stage 21 (early somitogenesis), stage 25 (initiation of heartbeat), stage 31 (late organogenesis), and stage 35 (pre-hatching). The PAH-content of unexposed embryos of both populations at the two-cell stage (Fig 1C) is comparable, except for naphthalenes, initially present in slightly higher concentrations among resistant embryos, suggesting maternal transfer of PAHs in the Elizabeth River resistant fish (Fig 1C).

Notably, the dry weight of resistant embryos is, on average, 21% less than that of sensitive embryos, and the lowest PAH concentrations inside the chorion occur among sensitive embryos exposed to clean extracts. The highest PAH concentrations are detected among resistant embryos exposed to polluted extracts (Fig 1C). These higher concentrations suggest additive effects of PAHs due to both maternal transfer and environmental exposure. However, PAH concentrations increase during development among sensitive embryos exposed to polluted extract (CP) when compared to resistant embryos (PP), possibly reflecting differences in the embryo's ability to detoxify and eliminate PAHs post complete organodifferentiation (stage 31). By stage 31, the liver, which expresses cytochrome P450 family enzymes with a crucial role in PAH detoxification [33], is functional. Importantly, all the PAHs detected in our embryo chemistry analyses are low-molecular weight, which was expected because the chorion plays a critical protective role from mechanical and chemical injury during teleost development. It is composed of dense multi-layer protein sheets, which are likely impermeable to high molecular-weight molecules during early development [49]. As embryos grow and reach advanced developmental stages during which the demand for the oxygen increases, the chorion becomes thinner, and its permeability increases.

We noted significant survival rate differences among sensitive and resistant embryos exposed to polluted sediment extracts (Fig 2A, ANOVA, $p < 0.05$). Sensitive embryos developing in clean sediment extracts (CC) up to stages 31 and 35 had the highest survival rates ($> 90\%$). Predictably, exposure of sensitive embryos to polluted sediment extracts (CP) resulted in significantly lower survival rates than among sensitive embryos developing in clean sediment extracts (CC); exposure to polluted sediment extracts did not significantly decrease survival rates among resistant embryos (PP) relative to resistant embryos developing in clean sediment extracts. Notably, there is lower survival yet a lack of measured anatomical and physiological response to the polluted sediments among resistant embryos exposed to polluted sediment extracts, while the sensitive population has a significant decline in survival. These data suggest that the resistant embryos are less likely to survive but are transiently unaffected by exposure to PAH-polluted sediment extracts. None of the sensitive embryos exposed to polluted sediment extracts hatched, even though $> 60\%$ of them survived until hatching, suggesting direct effects of PAH-polluted sediment extract on their survival. This 100% mortality among sensitive embryos (CP) post-hatch is characterized by significant developmental delays, reduced heart rates, and severe-to-extreme degrees of morphological abnormalities, relative to other treatment groups (Figs 2B–3B).

These significant developmental delays can confound gene expression differences due to developmental stage with those due to treatment when time (e.g., hours post-fertilization) is used to stage embryos. As evident in Fig 2C, b and Fig 2C, c, if one compares CP and CC embryo gene expression using time as the developmental reference, 74 genes have >2-fold expression differences among CP31 embryos and 32 genes have >2-fold expression differences in CC35 embryos; in these comparisons the effect of developmental stage on gene expression is confounded with that of sediment exposure, leading to false positives. In contrast, if one compares CP and CC embryos using anatomical and morphological similarities as the developmental reference, 49 genes have >2-fold expression differences among CP35 embryos and 9 genes have >2-fold expression differences in CC35 embryos, highlighting a more similar expression pattern due to the embryos being at the same developmental stage. In this comparison, gene differences only reflect the effect of sediment exposure. Our gene expression analyses use anatomically and morphologically staged embryos to distinguish treatment effects from developmental stage effects.

Assessment of heart morphology ranks all the sensitive embryos exposed to polluted sediment extracts as either severely or extremely deformed prior to hatching (Fig 3B). The degree of deformity is most evident in cardiac structure and function abnormalities, which are characterized by incomplete heart chamber differentiation (Fig 3A) and reduced contractile force and stroke volume, causing an inadequate supply of oxygenated blood to tissues during metabolically demanding developmental stages. While sensitive embryos from clean sediments (CC) and polluted embryos from both exposures (PC and PP) show no signs of PAH-induced heart damage, the sensitive embryos' heart rates are significantly lower (Fig 2D). Consequently, the sensitive embryos are, on average, three stages behind in their development prior to hatching (Fig 2B and 2C), and their extreme morphological abnormalities are particularly apparent in cardiac and cranio-facial structures (Fig 3A). Consequently, by stage 35 the heart resembles a barely-contracting thin transparent tube with clear fluid slowly trickling through, a condition known as a "tube heart" [50]. These embryotoxic alterations primarily associated with cardiac structure and function are like effects seen in zebrafish *(Danio rerio)* embryos exposed to a complex PAH mixture from crude oil [51]. The primary mechanism of embryo toxicity involves irreversible inhibition of cardiac conduction, subsequent loss of circulation and accumulation of edema, ultimately resulting in sublethal effects [48]. In contrast to the sensitive embryos exposed to polluted sediment extract, the sensitive embryos exposed to clean sediment extract and resistant embryos developing in both clean and polluted sediment extracts show fully formed, 2-chambered, healthy hearts forcefully pumping blood throughout an apparently normally developed embryo.

The relationships between developmental delays, cardiac function, and morphological abnormalities are evident in sensitive embryos exposed to polluted sediment extract. On average, the correlation between morphological alterations, heart rate, and developmental delays is 75% while less than a 30% correlation exists among these traits in the resistant embryo population (Fig 3C, ai). As sensitive embryos become more deformed and their morphology scores increase, their heart rates slow down because the deformed heart is not able to deliver enough blood to growing tissues ($R^2$ = 0.78, Fig 3C, bi). The morphological deformities, primarily affecting cardiac physiology, cause the same embryos to lag significantly in their development ($R^2$ = 0.82, Fig 3C, bi; $R^2$ developmental delays–heart rate = 0.65, Fig 3C, ci), being on average 3 stages (48 hours) behind, relative to other treatment groups (Fig 2C).

Histopathological analysis of late organogenesis (stage 31) embryos confirmed the morphological evidence for PAH induced toxicity. Deformities occur among sensitive embryos exposed to polluted sediment extracts but not among any of the other treatment groups (Fig 3D). Like the morphological analyses, the most convincing evidence of PAH-induced embryo

toxicity is abnormal cardiac tissue: deformed hearts, shaped as thin transparent tubes, are evident among 70% of the sensitive embryos exposed to PAH sediment extracts. Signs of embryo toxicity are fluid accumulation in the ventricular walls, subendothelial separation resulting in the effusion, loss of cardiac chamber wall integrity, myocyte vacuolation, and yolk vein hemorrhaging (Fig 3D, ci). Evidence of liver damage in the form of vacuolization and lipid accumulation within hepatocytes and kidney damage seen as swelling of epithelia and compromised tubular lumen was noted among 40% of the sensitive embryos exposed to polluted sediment extracts (Fig 3D, a and b). Morphological and histopathological analyses strongly suggest that the deformed hearts of sensitive embryos are not able to support the growth and required function of the vital organs and that the combined effects on liver and kidney lead to inability of these embryos to cope with the PAH toxicity. The cumulative effect of PAH toxicity results in failure to hatch and 100% mortality among sensitive embryos. Importantly, although the resistant embryos exposed to the same polluted extract show higher PAH concentrations during development (Fig 1C), their development is not drastically altered. They ultimately hatch and successfully complete development to free-swimming *F. heteroclitus*.

To better understand differences between the sensitive and resistant embryos at the molecular level, we measured gene expression for 6,789 *F. heteroclitus* genes during the same four developmental stages (stages 21, 25, 31, and 35). In total, 1178 genes were significantly differently expressed (17.4%, p < 0.01, mixed model ANOVA). 367 genes differed due to stage, 85 genes due to population, and 270 genes due to a treatment effect (Table 2). In 2-way interactions, expression of 108 genes were significant due to populations-by-stage; 118 genes were due to stage-by-treatment, and 431 genes were due to population-by-treatment effects. Hierarchical clustering of the 114 genes with a significant population-by-stage-by-treatment effects reveals a striking upregulation of a subset of genes only in stage 35 resistant embryos treated with polluted sediment extracts (PP35, Fig 3D). This subset is made up of 13 genes, including the 60S ribosomal subunit L22, parvalbumin beta, the serine protease inhibitor A3M, arsenite resistance protein, troponin C, leukocyte elastase inhibitor, trifunctional purine biosynthetic protein adenosine-3, and 6 unannotated genes. Expression of these genes is from 1.4 to 10.4-fold higher in these PP35 embryos than in any other embryo-treatment combination. Because resistant embryos treated with polluted sediment extract show few morphological effects of pollutant exposure and hatch normally, this subset of upregulated genes might be important for coping with the exposure.

Because we were more interested in treatment and population effects than developmental stage effects, we analyzed gene expression patterns at each developmental stage separately. All four stages show similar numbers of significantly differently expressed genes due to population, treatment, and population-by-treatment interactions except stage 25, which has few genes (21) significantly differently expressed with treatment, and stage 35, which has many genes (232) significantly differently expressed with treatment (compared to 93 and 157 genes for stages 21 and 31 respectively); stage 35 also has more genes significantly differently expressed with population compared to the other stages (83 genes vs. 28, 24, and 72 genes, Table 3).

One explanation for the low number of significantly differently expressed genes due to treatment at stage 25 is developmental canalization of gene expression so that treatment has little effect on gene expression. Gene expression canalization might occur during such critical developmental hallmarks as heartbeat initiation, which occurs at stage 25. In contrast, at stage 35, population and treatment effects on gene expression dominate: 83 genes have significant gene expression differences due to population and 232 genes are significantly differently expressed due to treatment. This avalanche of altered gene expression is reflected in the

chemical, morphological, and physiological results where stage 35 sensitive embryos have the greatest pollutant burdens and show the most severe effects.

Hierarchical clustering of significant genes at each stage (Fig 4B) shows that at stages 21, 31, and 35, the most similar gene expression patterns are based on treatment so that sensitive and resistant embryos developing in clean sediment extracts cluster together (PP and CP) and sensitive and resistant embryos developing in polluted extracts cluster together (PC and CC). This is not the case for stage 25 embryos, where resistant embryos exposed to clean sediment and sensitive embryos exposed to polluted sediment (PC and CP) share similarities in gene expression patterns as do polluted embryos exposed to polluted sediment and sensitive embryos exposed to clean sediment extracts (PP and CC). Importantly, the most significantly differently expressed genes (308) and the most dramatic gene expression pattern differences based on treatment effects occur at stage 35: both sensitive and resistant embryos developing in clean sediment extracts (CC and PC) cluster together and show striking gene expression similarities prior to hatching, with less variation in their expression profiles compared to the CP and PP cluster. This is supported with our embryo chemistry, developmental delay assessments, and physiology and morphology data, showing that the most distinct differences between sensitive and resistant populations caused by cumulative treatment effects are at the pre-hatching stage (stage 35).

Many of the 308 genes (Fig 4B, d) are clustered within five main functionally relevant gene clusters. We use fold change difference in gene expression between CC and CP, PC, or PP embryos to illustrate relative treatment effects. In Cluster 1, seventy-two genes are upregulated in sensitive embryos developing in polluted sediment extracts (CP) compared to the other treatments. Upregulation of five ribosomal subunit assembly genes in CP vs. CC embryos (28s, 40s, and 60s; 2.09–3.61-fold) and glyceraldehyde-3-phosphate dehydrogenase (GAPDH; 3.31-fold) may be indicative of response to DNA damage and tumor suppression [52], although the intermediate p53 induction in this pathway was not observed. Colonic and hepatic tumor over-expression gene 14 (CH-TOG; 1.80-fold) induction may also suggest active anti-tumor response, while the induction (2.12-fold) of superoxide dismutase (SOD), which plays a role in scavenging free radicals during oxidative stress [34,53] and helps eliminate reactive oxygen species, free radicals, and reactive intermediates sometimes produced during PAH metabolism [47], is indicative of oxidative stress response.

Within Cluster 1, 26 genes are only induced in stage 35 sensitive embryos treated with polluted sediment extract (Fig 4B, d). These genes are refractory to induction in resistant embryos treated with polluted sediment extract. One of these genes, CYP1A, has previously been shown to be refractory to induction by typical CYP1A inducers such as PAHs not only in Elizabeth River embryos, but also in *F. heteroclitus* from two other highly polluted sites, New Bedford Harbor, MA and Newark Bay, NJ [23,54]. Similarly, we report significant upregulation of CYP1A in CP embryos (5.9-fold, compared to CC) and slight downregulations in resistant embryos (0.78-fold, PP vs. CC; 0.93-fold, PC vs. CC). CYP1A is a phase I enzyme in PAH detoxification mechanisms, and the lack of induction to typical inducers such as PAHs has been postulated to be a protective mechanism as reduced ligand-binding does not activate receptor-mediated toxicity responses in resistant fish [55]. Similar lack of induction observed in the remaining 25 genes might be part of the adaptive signature among polluted embryos exposed to polluted sediment extract.

In Cluster 2 (Fig 4B, d), twenty-seven genes are downregulated among resistant embryos in polluted extracts and resistant embryos developing in clean water (PP and PC) as compared to sensitive embryos. This gene expression profile overlap is important because similarly regulated gene expression may be indicative of adaptive responses and fixed gene expression regardless of exposure in the resistant population. Gene functions include regulation of cell

division (2 genes), hemoglobin (2 genes) and insulin (1 gene) synthesis, metabolism (1 gene) and protein degradation (1 gene). Reduced expression of cell division control protein 2 (0.49-fold, PC vs. CC), programmed cell death 10 (0.74-fold, PC vs. CC), and proteasome sub-unit alpha type-3 (0.56-fold, PC vs. CC) [56–58], suggest that cell proliferation is favored with reduced recycling to maintain tissue integrity, as resources are limited, and excessive apoptosis is likely to be induced in a polluted environment. The cost of adaptation to polluted environments also may be evident in downregulation of both hemoglobin chains, alpha (0.66-fold, PC vs. CC) and beta-2 (0.57-fold, PC vs. CC), leading to reduced hemoglobin synthesis and an anemic phenotype marked by decreased ability to oxygenate tissues and remove excess $CO_2$, hence compromising metabolic output. Such effects, combined with downregulation of insulin precursor (0.68-fold, PC vs. CC) and dehydrogenase/reductase SDR family member IV (0.81-fold, PC vs. CC), may result in reduced systemic metabolic effects.

In Cluster 3, high relative expression among PP embryos is evident in genes associated with skeletal muscle contraction, which include actin (3.37-fold), parvalbumin-beta (5.32-fold), tropomodulin 1 (1.93-fold) and 4 (2.74-fold), troponin C (4.26-fold) and T (3.57-fold), myosin heavy chain (5.42-fold), and titin (4.51-fold). The argument that toxic exposure results in the higher metabolic demand is supported by upregulation of critical enzymes in cellular respiration–adenylate kinase (2.66-fold), pyruvate kinase (3.14-fold), NAD(P) transhydrogenase (2.45-fold), NADH-ubiquinone oxidoreductase (5.37-fold), and ATP synthase (1.70-fold), since the increased muscle activity, possibly induced by PAH exposure and oxidative stress [59], demands higher levels of cellular ATP utilization.

In Cluster 4, sixty-nine genes are downregulated among resistant and sensitive embryos in polluted extracts (PP and CP). Some of the gene downregulation is indicative of PAH-induced physiological stress, including altered cardiac function and clotting, slower metabolism, compromised cellular immune response, and detoxification mechanism. Downregulation of genes associated with cardiac function edema and blood clotting–atrial natriuretic factor precursor (0.54-fold), myosin regulatory light chain 2 smooth muscle isoform (0.52-fold), heparin cofactor 2 precursor (0.51-fold), Von Willebrand factor precursor (0.75 fold), and sodium/potassium-transporting ATPase (0.57-fold)–correlate with PAH-induced cardiomyopathy, pericardial edema, and dysregulation of blood clotting mechanisms, all evident in sensitive embryos exposed to polluted sediment extract (Fig 3A). Likewise, toxic effects during development may involve downregulation of genes associated with a delicate balance between cell proliferation/differentiation, apoptosis, and cell migration during embryogenesis: inositol polyphosphate-4-phosphatase (0.41-fold), Ras (0.57-fold), atrophin-1 (0.79-fold), and ARF GTPase-activating protein GIT2 (0.44-fold). Downregulation of these genes suggests decreased cellular turnover, tissue remodeling and cell migration inhibition [60–63].

Lysozymes are integral innate immunity enzymes, capable of degrading bacterial cells wall and catabolizing cellular components [64]. Downregulation of three lysozyme genes, lysozyme C (0.51-fold), G (0.46-fold), and lysosomal acid phosphatase (0.58-fold) among CP and PP embryos suggests compromised immunocompetence due to toxicant exposure. Moreover, downregulation (0.46-fold) of UDP-glucuronosyltransferase (UDPGT) among CP and PP embryos is critical because this enzyme catalyzes the transfer of glucuronic acid to hydrophobic moiety during Phase II detoxification [65]. Among sensitive embryos exposed to PAH-polluted sediment extracts, upregulation of CYP1A and downregulation of UDPGT results in accumulation of toxic polar secondary metabolites eliciting toxic effects. However, as CYP1A is downregulated among PP embryos, the PAH parent compound would not be activated, and UDPGT downregulation should not induce the dramatic hallmarks of PAH toxicity evident among sensitive embryos exposed to polluted sediment extracts.

Notably, Cluster 5 does not show a clear gene expression pattern relative to treatment, but there are several genes whose expression might indicate compensatory mechanism. The upregulation of NADH-ubiquinone (1.75-fold, PC vs. CC) and putative oxidoreductase (1.76-fold, PC vs. CC) in PP and PC embryos (Fig 4B, d-Cluster 5) could counteract the reduced metabolic activities signified in Cluster 2. Increased expression of UDP-glucuronic acid/UDP-N-acetylgalactosamine transporter (1.26-fold, PC vs. CC), which plays a role in biosynthesis of a chondroitin sulfate, a compound necessary for formation of 16 cartilage extra-cellular matrix and skeletal development [66], may also be complementary to upregulation of skeletal muscle movement genes (Fig 4B, d-Cluster 3) among resistant embryos developing in polluted sediment extracts.

## Conclusion

Our study reveals important differences between sensitive and resistant embryo populations exposed to contaminated PAH sediment extracts. Chemistry, physiology, *in vivo* morphology, and histopathology results clearly show that the PAH-contaminated sediments are significantly more toxic to sensitive embryos from a clean environment than to resistant embryos from in a highly contaminated environment. Gene expression data during four critical developmental stages reveal genes whose expression significantly differs between these embryo populations and whose expression patterns at each developmental stage may clarify mechanisms of sensitivity and resistance to polluted environments during animal development. While expression patterns of selected significant genes highlight important differences between sensitive and resistant embryos, they also confirm the complexity of interactions associated with mechanisms of sensitivity and resistance within developing organisms.

## Supporting information

**S1 Table. List of polycyclic aromatic hydrocarbons (PAHs) tested in the study.** The table includes chemical analysis data for both sediment and passive sampling devices (PSDs). (XLSX)

## Acknowledgments

We thank Liisa Bozinovic for gene expression data quality control, Douglas L. Crawford for critical discussions and comments on the manuscript and for help printing microarrays, and UCSD Extension for providing statistical software for data analysis.

## Author Contributions

**Conceptualization:** Goran Bozinovic, David Hinton, Marjorie F. Oleksiak.

**Data curation:** Goran Bozinovic, Damian Shea, Zuying Feng, Marjorie F. Oleksiak.

**Formal analysis:** Goran Bozinovic, Damian Shea, Zuying Feng, Marjorie F. Oleksiak.

**Funding acquisition:** Marjorie F. Oleksiak.

**Investigation:** Goran Bozinovic, Damian Shea, David Hinton, Tim Sit, Marjorie F. Oleksiak.

**Methodology:** Goran Bozinovic, Damian Shea, David Hinton, Tim Sit, Marjorie F. Oleksiak.

**Project administration:** Goran Bozinovic, Marjorie F. Oleksiak.

**Resources:** Damian Shea, Marjorie F. Oleksiak.

**Supervision:** Damian Shea, David Hinton, Tim Sit, Marjorie F. Oleksiak.

**Validation:** Goran Bozinovic, Damian Shea, Zuying Feng, Tim Sit, Marjorie F. Oleksiak.

**Visualization:** Goran Bozinovic, Zuying Feng, David Hinton, Tim Sit, Marjorie F. Oleksiak.

**Writing – original draft:** Goran Bozinovic, Zuying Feng, David Hinton, Marjorie F. Oleksiak.

**Writing – review & editing:** Goran Bozinovic, Damian Shea, Zuying Feng, David Hinton, Tim Sit, Marjorie F. Oleksiak.

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
