## [Decision Letter · Decision Letter 0]

24 Feb 2021

PONE-D-21-02056

Pollution effects on sensitive and resistant embryos: Integrating structure and function with gene expression

PLOS ONE

Dear Dr. Bozinovic,

Thank you for submitting your manuscript to PLOS ONE. After careful consideration, we feel that it has merit but does not fully meet PLOS ONE’s publication criteria as it currently stands. Therefore, we invite you to submit a revised version of the manuscript that addresses the points raised during the review process.

We look forward to receiving your revised manuscript.

Kind regards,

Yi Cao

Academic Editor

PLOS ONE

Journal Requirements:

3. In your Methods section, please include a comment about the state of the animals following this research. Were they released, euthanized or housed for use in further research? If any animals were sacrificed by the authors, please include the method of euthanasia and describe any efforts that were undertaken to reduce animal suffering.

4. We note that you are reporting an analysis of a microarray, next-generation sequencing, or deep sequencing data set. PLOS requires that authors comply with field-specific standards for preparation, recording, and deposition of data in repositories appropriate to their field. Please upload these data to a stable, public repository (such as ArrayExpress, Gene Expression Omnibus (GEO), DNA Data Bank of Japan (DDBJ), NCBI GenBank, NCBI Sequence Read Archive, or EMBL Nucleotide Sequence Database (ENA)). In your revised cover letter, please provide the relevant accession numbers that may be used to access these data. For a full list of recommended repositories, see http://journals.plos.org/plosone/s/data-availability#loc-omics or http://journals.plos.org/plosone/s/data-availability#loc-sequencing.

5. Thank you for including your ethics statement:  "All experiments were performed according to approved protocols per Institutional Animal Care and Use Committees, North Carolina State University and Duke University.".   

Please amend your current ethics statement to confirm that your named ethics committee specifically approved this study.

For additional information about PLOS ONE submissions requirements for ethics oversight of animal work, please refer to http://journals.plos.org/plosone/s/submission-guidelines#loc-animal-research  

Reviewers' comments:

Reviewer's Responses to Questions

**Comments to the Author**

1. Is the manuscript technically sound, and do the data support the conclusions?

Reviewer #1: Yes

Reviewer #2: Yes

2. Has the statistical analysis been performed appropriately and rigorously? 

Reviewer #1: Yes

Reviewer #2: No

3. Have the authors made all data underlying the findings in their manuscript fully available?

Reviewer #1: Yes

Reviewer #2: Yes

4. Is the manuscript presented in an intelligible fashion and written in standard English?

Reviewer #1: Yes

Reviewer #2: Yes

5. Review Comments to the Author

Reviewer #1: In this manuscript, the author chose the F. heteroclitus embryos from the Elizabeth River Superfund population and the relatively unpolluted population nearby to simulate the sensitive and resistant of the natural population, and exposed them to contaminated and clean sediment extracts to explore the mechanism of POP population differences through the detection of morphology, physiology, chemical absorption and gene expression changes of embryos at different key developmental stages. Starting from developmental toxicity, this article combined with actual environmental exposure, which is an interesting and meaningful study.

The article is well written. However, the explanation of the exposure dose of the extract in this study is not very clear. Is the composition of the extract at different times or seasons consistent? Explaining these clearly helps prove the significance of the research.

Regarding the follow-up of microarray, the author is hoped to add the results verification of the differentially expressed genes, which would appear more solid and credible.

Reviewer #2: The present study chemically characterized clean and heavily polluted sites and exposed fish embryos to PAH polluted sediment extracts during four critical developmental stages, and used gene expression analysis for different developmental to identify differentially expressed genes which helps to explain mechanisms of sensitivity and resistance to polluted environments during vertebrate animal development. The manuscript is generally well prepared and the results are informative, and I have some comments to improve the ms based on its present form.

Title: the primary pollutants investigated in the study are polycyclic aromatic hydrocarbons, which should be reflected in the title. Currently the title does not identify what pollutants are the pollutants of interest.

The results of gene expression should consider the issue of multiple testing adjustment. It is possible that many of the identified genes are just false positives simply because of the high number of genes (n=6769) being tested. May consider False Discovery Rate, Bonferroni correction or other similar methods.

It is unclear how many kinds of PAHs were tested in the study, and would better provide a list of analyzed individual PAHs in the methods section (along with their molecular weights and low/high molecular weight designation), either in text or a supplemental table to facilitate the understanding.

Fig 1. The figure legend explained that “Low concentrations of only 10 of the 42 PAHs tested, mostly low molecular weight naphthalenes, are detected at the reference site”, however, the B panel showed no blue columns for the reference site. Please check.

Fig 2. Please indicate what is the meaning of the asterisk as shown in the figure. There is no explanation for the asterisk in the legend. Also be consistent to use ‘one-way ANOVA’ and ‘1-way ANOVA’ and related p values for cutoff (<0.05 vs. <0.01).

Fig 3. The issue of asterisk also exists here. Please explain the meaning of R2 in the C panel and how to calculate it, also explain the meaning of the green curves in the same panel.

6. PLOS authors have the option to publish the peer review history of their article (what does this mean?). If published, this will include your full peer review and any attached files.

Reviewer #1: No

Reviewer #2: No

---

## [Author Response · Author response to Decision Letter 0]

4 Mar 2021

Thank you for your valuable comments and suggestions which strengthen the content and relevance of our manuscript. Please consider our responses below.

Comments to the Author

1. Is the manuscript technically sound, and do the data support the conclusions?

Reviewer #1: Yes

Reviewer #2: Yes

2. Has the statistical analysis been performed appropriately and rigorously? 

Reviewer #1: Yes

Reviewer #2: No

3. Have the authors made all data underlying the findings in their manuscript fully available?

Reviewer #1: Yes

Reviewer #2: Yes

4. Is the manuscript presented in an intelligible fashion and written in standard English?

Reviewer #1: Yes

Reviewer #2: Yes

5. Review Comments to the Author

Reviewer #1: In this manuscript, the author chose the F. heteroclitus embryos from the Elizabeth River Superfund population and the relatively unpolluted population nearby to simulate the sensitive and resistant of the natural population, and exposed them to contaminated and clean sediment extracts to explore the mechanism of POP population differences through the detection of morphology, physiology, chemical absorption and gene expression changes of embryos at different key developmental stages. Starting from developmental toxicity, this article combined with actual environmental exposure, which is an interesting and meaningful study.

The article is well written. However, the explanation of the exposure dose of the extract in this study is not very clear. Is the composition of the extract at different times or seasons consistent? Explaining these clearly helps prove the significance of the research.

Since all sediment samples were collected in April 2007 during a single sampling event, the variation in dosing is not addressed. We now clearly state the single-season sampling event (line 97-100); relevant sediment chemical composition is presented in Figure 1A-B, and we added a supplementary table (S1 Table) that lists all PAHs tested and corresponding concentrations; body burdens of exposed embryos are shown in Figure 1C. 

Regarding the follow-up of microarray, the author is hoped to add the results verification of the differentially expressed genes, which would appear more solid and credible.

We appreciate the comment about verifying the microarray data with qRT-PCR assay. There is, however, an experimental justification in certain instances that indicate such validation is not necessary. Since biological replication and reproducibility of results is the aim of the microarray experimental design, a well-balanced design with sufficient biological replication for array data analyses yields a statistical power for differential gene expression detection. Such microarray platforms with rigorous statistical analysis are sufficient to not warrant technical replicates. Moreover, if the same samples are analyzed twice but only with a different technology (microarray vs. qRT-PCR), the results would serve as a technical validation, possibly revealing a "technology bias". Lastly, microarrays vs. qRT-PCR may be relevant when the “true” fold-changes are a desired outcome, and the arrays were used as a screening tool. Analyzing different samples with the same technique will provide an estimate of the biological and technical variance. Analyzing different samples with a different technique will reveal a combination of "technology bias" and the "between-assay variance". Our study focuses on identifying differential expression, while accurate measurement of fold-changes, which is required for setting up quantitative dynamic models of gene-regulation or other molecular-physiology processes, is not the goal of our study. We respectfully include three references addressing the questions of necessity of microarray data validation relative to sufficient biological replication and balanced experimental design.

Chuaqui, R. F., Bonner, R. F., Best, C. J., Gillespie, J. W., Flaig, M. J., Hewitt, S. M., ... & Emmert-Buck, M. R. (2002). Post-analysis follow-up and validation of microarray experiments. Nature genetics, 32(4), 509-514.

Morey, J. S., Ryan, J. C., & Van Dolah, F. M. (2006). Microarray validation: factors influencing correlation between oligonucleotide microarrays and real-time PCR. Biological procedures online, 8(1), 175-193.

Wise, R. P., Moscou, M. J., Bogdanove, A. J., & Whitham, S. A. (2007). Transcript profiling in host–pathogen interactions. Annu. Rev. Phytopathol., 45, 329-369.

Reviewer #2: The present study chemically characterized clean and heavily polluted sites and exposed fish embryos to PAH polluted sediment extracts during four critical developmental stages, and used gene expression analysis for different developmental to identify differentially expressed genes which helps to explain mechanisms of sensitivity and resistance to polluted environments during vertebrate animal development. The manuscript is generally well prepared and the results are informative, and I have some comments to improve the ms based on its present form.

Title: the primary pollutants investigated in the study are polycyclic aromatic hydrocarbons, which should be reflected in the title. Currently the title does not identify what pollutants are the pollutants of interest.

We have added “PAH” to the title to reflect the focus of the study. The title now reads “PAH-pollution effects on sensitive and resistant embryos: Integrating structure and function with gene expression” (line 3).

The results of gene expression should consider the issue of multiple testing adjustment. It is possible that many of the identified genes are just false positives simply because of the high number of genes (n=6769) being tested. May consider False Discovery Rate, Bonferroni correction or other similar methods.

We agree with the comment and updated Table 2 to show the number of significant genes after False Discovery Rate filtration (pFDR < 0.05) is applied. We modified the manuscript accordingly to address the false positive issue in line 602-605 and 615-616. We also reinforced the original statement in the Methods “Using this p-value reveals more genes that may be differentially expressed but risks identifying genes that may be false positives.” (line 351-352) with “False discovery rate (pFDR < 0.05) was also used for filtering residuals.” (line 352-353)

It is unclear how many kinds of PAHs were tested in the study, and would better provide a list of analyzed individual PAHs in the methods section (along with their molecular weights and low/high molecular weight designation), either in text or a supplemental table to facilitate the understanding.

We appreciate the comment and now list all PAHs tested in the study, including their molecular weights, in S1 Table. We also clarified the designation of low (fewer than 4 aromatic rings) and high (4 or more aromatic rings) molecular weight PAHs (line 359-361). 

Fig 1. The figure legend explained that “Low concentrations of only 10 of the 42 PAHs tested, mostly low molecular weight naphthalenes, are detected at the reference site”, however, the B panel showed no blue columns for the reference site. Please check.

For Fig 1B, the concentrations of PAHs detected at the reference site are significantly lower (non-zero concentrations are mostly below 10 ng/L) than those for Elizabeth River and are thus not visible on the figure. We have adjusted the axis so the PAH with the highest concentration (C1 - Naphthalenes, 11.86 ng/L) is more visible. This issue is also addressed in line 370-373 and 386-387 and further detailed in S1 table. 

Fig 2. Please indicate what is the meaning of the asterisk as shown in the figure. There is no explanation for the asterisk in the legend. Also be consistent to use ‘one-way ANOVA’ and ‘1-way ANOVA’ and related p values for cutoff (<0.05 vs. <0.01).

We much appreciate this comment and have modified the figure so that the statistical significance is more clearly illustrated. We now explain the meaning of the asterisk in line 438-439 (“*, p < 0.05; **, p < 0.01; ***, p < 0.001; ****, p < 0.0001.”). We also changed “1-way ANOVA” to “one-way ANOVA” (line 438).

Fig 3. The issue of asterisk also exists here. Please explain the meaning of R2 in the C panel and how to calculate it, also explain the meaning of the green curves in the same panel.

We have modified the figure so that the statistical significance is more clearly illustrated and now explain the meaning of the asterisk in line 496 (“****, p < 0.0001.”). The R2 and regression line are now explained in line 495-497.

---

## [Decision Letter · Decision Letter 1]

18 Mar 2021

PAH-pollution effects on sensitive and resistant embryos: Integrating structure and function with gene expression

PONE-D-21-02056R1

Dear Dr. Bozinovic,

We’re pleased to inform you that your manuscript has been judged scientifically suitable for publication and will be formally accepted for publication once it meets all outstanding technical requirements.

Kind regards,

Yi Cao

Academic Editor

PLOS ONE

Additional Editor Comments (optional):

Reviewers' comments:

Reviewer's Responses to Questions

**Comments to the Author**

1. If the authors have adequately addressed your comments raised in a previous round of review and you feel that this manuscript is now acceptable for publication, you may indicate that here to bypass the “Comments to the Author” section, enter your conflict of interest statement in the “Confidential to Editor” section, and submit your "Accept" recommendation.

Reviewer #1: All comments have been addressed

Reviewer #2: All comments have been addressed

2. Is the manuscript technically sound, and do the data support the conclusions?

Reviewer #1: Yes

Reviewer #2: Yes

3. Has the statistical analysis been performed appropriately and rigorously? 

Reviewer #1: Yes

Reviewer #2: Yes

4. Have the authors made all data underlying the findings in their manuscript fully available?

Reviewer #1: Yes

Reviewer #2: Yes

5. Is the manuscript presented in an intelligible fashion and written in standard English?

Reviewer #1: Yes

Reviewer #2: Yes

6. Review Comments to the Author

Reviewer #1: (No Response)

Reviewer #2: My comments have been addressed appropriately by the authors, and I have no further comments on the revised manuscript.

7. PLOS authors have the option to publish the peer review history of their article (what does this mean?). If published, this will include your full peer review and any attached files.

Reviewer #1: No

Reviewer #2: No

---

## [Editor Report · Acceptance letter]

22 Mar 2021

PONE-D-21-02056R1 

PAH-pollution effects on sensitive and resistant embryos: Integrating structure and function with gene expression 

Dear Dr. Bozinovic:

I'm pleased to inform you that your manuscript has been deemed suitable for publication in PLOS ONE. Congratulations! Your manuscript is now with our production department. 

Kind regards, 

on behalf of

Dr. Yi Cao 

Academic Editor

PLOS ONE